# COMPOSITIONAL CHOKE POINTS: EVIDENCE OF EMERGENT RANK COLLAPSE LIMITING ITERATIVE GENERALIZATION

## ABSTRACT

Neural networks excel at interpolation but struggle with compositional generalization. This paper constructs a deterministic benchmark where models must learn a symbolic cyclic permutation rule $\pi^{(n)}(s)$ from stochastically rendered images. While a hybrid symbolic–neural oracle achieves perfect extrapolation, standard feedforward and transformer networks fail catastrophically beyond the training distribution, performing at chance. Representational analysis reveals a collapse in effective rank, consistent with a memorization-dominated solution rather than algorithmic learning. These results demonstrate a fundamental limitation in the inductive biases of current neural architectures for systematic reasoning.

## 1 INTRODUCTION

Deep neural networks have achieved remarkable empirical success across perception, language, and scientific domains. This limitation has been documented across tasks requiring rule reuse, symbolic manipulation, or iterative computation (Bahdanau et al., 2018; Lin et al., 2023; Pastore & Carnini, 2021). In these settings, strong in-distribution performance frequently collapses to chance-level accuracy under extrapolation, raising fundamental questions about what has been learned. These failures are commonly attributed to inductive biases induced by gradient-based optimization, which favor interpolative solutions over representations that encode reusable algorithmic structure (Bonnasse-Gahot, 2022; Mansour et al., 2019; Smith et al., 2021). As a result, networks may rely on memorization or local interpolation while failing to acquire reusable algorithmic structure.

Persistent failures of compositional generalization have been observed across modalities, including vision, language, and reinforcement learning, even as model and dataset scale increase (Lin et al., 2023; Azizi et al., 2023). Controlled evaluation frameworks further show that benchmark performance does not imply systematic generalization, motivating diagnostic tasks that explicitly probe extrapolation and rule learning (Camposampiero et al., 2025). Architecturally, hybrid and symbolic approaches demonstrate improved generalization by injecting algebraic or structural priors, particularly in abstract reasoning domains (Zhang et al., 2022). However, the representational mechanisms underlying these failures remain poorly understood. While most prior work emphasizes behavioral outcomes, less attention has been paid to the geometry of learned representations. Studies of representation structure reveal intrinsic dimensionality reduction and neural collapse in trained networks (Ansuini et al., 2019; Fang et al., 2021). Although such effects do not necessarily imply information loss in standard classification settings (Yang et al., 2023), they raise concerns about whether low-dimensional latent spaces can support iterative or stateful computation.

Related limitations appear in algorithmic reasoning. Transformers struggle with tasks requiring precise iteration or deep recursion (Sanford et al., 2024), while recurrent and iterative architectures exhibit optimization and stability pathologies (Neshatpour et al., 2019; Zucchet & Orvieto, 2024). Together, these findings suggest that the absence of inductive biases for state preservation and iteration constitutes a central obstacle to systematic reasoning (Goyal & Bengio, 2022).

In this work, we study the representational origins of compositional generalization failure using a controlled symbolic transition task that cleanly separates perceptual abstraction from algorithmic computation. By analyzing the effective rank of penultimate-layer activations, we identify an emergent representational choke point in which latent dimensionality collapses during optimization,

coinciding with catastrophic extrapolation failure. We introduce a deterministic, fully reproducible benchmark isolating symbolic iteration from perceptual complexity and show that standard feedforward and projection-and-mixing architectures default to non-reusable, interpolative representations despite the task being algorithmically solvable. Together, these results provide a mechanistic account of how inductive biases and representation geometry constrain systematic generalization in deep learning.

## 2 Experimental Methodology: A Controlled Test for Systematic Generalization

We construct a deterministic experimental framework to test whether neural networks can learn and generalize compositional symbolic rules while separating perceptual abstraction from algorithmic computation. All experiments are fully reproducible under a fixed global random seed.

The task is to compute a symbolic transition

$$\mathcal{T}(s, n) = \pi^{(n)}(s),$$

where $s \in \mathcal{S}$ is a discrete symbol with $|\mathcal{S}| = 8$ and $\pi : \mathcal{S} \to \mathcal{S}$ is a deterministic cyclic permutation. Given a perceptual rendering of $s$ and an iteration count $n$, the model must predict the final symbol $t = \pi^{(n)}(s)$.

To eliminate perceptual shortcuts, symbols are rendered via a stochastic **TextureRenderer**

$$\mathcal{R} : \mathcal{S} \to [0, 1]^{3 \times 16 \times 16},$$

in which only a symbol-specific noise component is invariant across renderings, forcing abstraction of symbolic identity from variable visual inputs.

Each sample is a tuple $(\mathbf{I}_s, \hat{n}, t)$, with $\mathbf{I}_s = \mathcal{R}(s)$ and $\hat{n} = n/100$. Training uses iteration depths $n \in \{1, \ldots, 10\}$, while testing spans $n \in \{1, \ldots, 100\}$, placing 90% of test samples outside the training regime. The training and test sets contain 10,000 and 2,000 samples, respectively.

We evaluate three architectures. A **Feedforward Network (FFN)** jointly learns perception and reasoning by concatenating image features with $\hat{n}$ and predicting $t$ via an MLP. A **Simplified Transformer** projects image features and $\hat{n}$ into a shared latent space prior to nonlinear mixing, testing whether modality projection alone enables systematic generalization. An **Oracle Module Network** separates perception from reasoning, identifying $s$ from $\mathbf{I}_s$ and computing $\mathcal{T}(s, n)$ deterministically, providing an upper bound on achievable performance.

All models are trained for 50 epochs using Adam. FFN and Transformer models optimize cross-entropy loss over $t$, while the Oracle optimizes over $s$. Representations are analyzed via penultimate-layer activations. We quantify latent geometry using the **effective representational rank**, where rank collapse indicates reliance on memorization rather than structured computation.

## 3 Empirical Results: Catastrophic Failure of Systematic Generalization

The results reveal a sharp dissociation between inductive learning and systematic generalization. While standard neural architectures fit the training distribution, they fail completely to extrapolate the underlying compositional rule.

**Training performance.** Both the Feedforward Network (FFN) and the Simplified Transformer successfully learned the training set. The FFN reached $100.0\%$ accuracy by epoch 35, while the Transformer converged to $98.35\%$ accuracy by epoch 50. The Oracle's perceptual module also achieved $100.0\%$ accuracy in identifying the initial symbol $s$ from its rendered image. These results confirm that the task is learnable in-distribution, optimization is effective, and perceptual obfuscation is not a limiting factor.

**Extrapolation failure and generalization cliff.** Despite strong training performance, both purely neural models performed at chance level during extrapolation. For an 8-way classification task

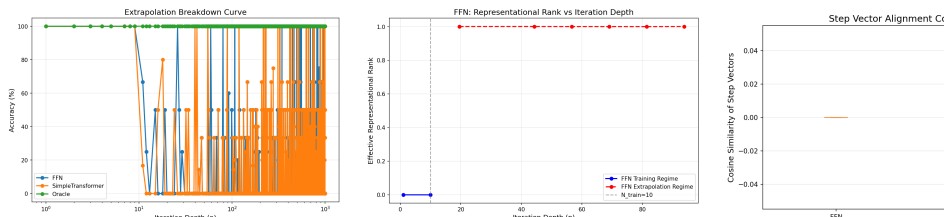

Figure 1: **Compositional choke point and representational failure. Left:** Accuracy collapses to chance immediately beyond the training regime. **Middle:** FFN representations exhibit severe rank collapse across iteration depths. **Right:** Representations conditioned on the iteration count $\hat{n}$ lack structured alignment, indicating non-algorithmic use of the iteration signal.

(chance $12.5\%$), the FFN achieved $12.90\%$ test accuracy and the Simplified Transformer $12.15\%$, both effectively at chance performance for an 8-way classification task. In contrast, the Oracle Module Network achieved $100.0\%$ test accuracy by executing the symbolic transition rule explicitly.

Because the Oracle shares the same perceptual backbone, its perfect extrapolation isolates the failure to the neural models' inability to acquire the compositional rule, with accuracy collapsing immediately beyond the training regime (Figure 1).

The FFN's mean accuracy on strictly extrapolative samples ($n > 10$) was $11.86\%$. Since $90\%$ of test samples fall in this regime, overall test performance is dominated by extrapolation failure.

**Representational analysis.** To identify the mechanistic source of failure, we analyzed FFN activations from the penultimate layer (pre_logits, $d = 128$). Activations exhibited extreme variance and poor conditioning, consistent with unstable latent geometry.

Effective representational rank provides the key diagnostic. For training samples ($n \leq 10$), the effective rank collapsed to approximately $R_{\text{eff}} \approx 1.00$, indicating a near-complete rank–1 degeneracy. For extrapolation samples ($n > 10$), the effective rank remained similarly low ($R_{\text{eff}} \approx 1.00$), confirming the absence of any high-dimensional latent structure supporting iteration. This collapse implies that representations for all training inputs are nearly identical, differing only by small numerical perturbations rather than structured variation aligned with the underlying symbolic transition.

Rather than encoding the rule $\pi$, the FFN implements a memorization strategy: a lookup-table-like mapping keyed by small variations in the penultimate activations. When presented with a novel iteration count, activations fall outside the memorized subspace, causing the final linear layer to produce effectively random predictions.

The Simplified Transformer exhibited more moderate activation statistics (mean $\approx 0.44$, std $\approx 32.78$), but its covariance spectrum was sharply dominated by a single eigenvalue, yielding an effective rank close to one and indicating a similar representational collapse.

Table 1: Training and extrapolation performance across models. Despite perfect or near-perfect training accuracy, purely neural models perform at chance under extrapolation, while the symbolic–neural Oracle generalizes perfectly.

| Model | $A_{\text{train}}$ | $A_{\text{test}}$ | $\Delta G$ |
|---|---|---|---|
| Feedforward Network (FFN) | 100.00% | 12.90% | 87.10% |
| Simplified Transformer | 98.35% | 12.15% | 86.20% |
| Oracle Module Network | 100.00% | 100.00% | 0.00% |
| Random Baseline | 12.50% | 12.50% | 0.00% |

# 4 INTERPRETATION: INDUCTIVE BIASES AND THE PATH FORWARD

These results highlight fundamental limitations in the inductive biases of standard neural architectures and clarify why systematic generalization remains difficult to achieve.

**The failure is algorithmic, not perceptual.** The Oracle Module Network's perfect performance establishes that perceptual abstraction is fully solvable by a standard CNN. Consequently, the extrapolation failure of the FFN and Simplified Transformer cannot be attributed to visual recognition difficulty. Instead, the failure is isolated to the reasoning component: the models do not acquire the cyclic permutation rule $\pi^{(n)}$ in a general, composable form and cannot apply it beyond the iteration depths observed during training. Importantly, the Oracle does not benefit from privileged perceptual information or additional supervision. It shares the identical convolutional perceptual backbone with the neural models and differs only in the explicit factorization of perception and symbolic rule execution. Its role is not to propose a trainable alternative but to establish an upper bound on achievable generalization when the correct algorithmic structure is available.

**Bias toward interpolation and memorization.** Rather than learning an internal mechanism that applies a transformation iteratively, the networks encode input–output correspondences directly. This strategy succeeds within the training support but fails completely outside it. The abrupt drop in performance at $n = N_{\text{train}} + 1$ constitutes a *compositional choke point* and serves as a clear diagnostic of non-algorithmic learning. If the models had learned an explicit procedure "identify $s$, then apply a shift operation $n$ times" performance would degrade gradually with increasing $n$. Instead, accuracy collapses immediately to chance, indicating reliance on conditional correlations rather than causal, compositional rules.

This behavior reflects an inductive bias shaped by gradient-based optimization in high-capacity continuous models, which favors low-norm interpolative solutions over compact algorithmic representations.

**Limits of modality projection.** The Simplified Transformer performs no better than the FFN, despite explicitly projecting image features and iteration count into a shared latent space. This shows that modality projection alone does not induce the inductive bias required for learning iterative or algebraic structure. In the absence of explicit mechanisms for state manipulation or iteration, the scalar $n$ functions only as an additional feature, not as a control signal.

**Representational analysis as a diagnostic.** Effective representational rank provides a mechanistic diagnostic beyond accuracy metrics. Rank collapse—particularly an effective rank approaching one in the training regime—indicates that the network fails to maintain a structured latent space and instead implements memorization. This analysis offers a principled criterion for distinguishing interpolative solutions from genuinely algorithmic ones and suggests a clear target for future architectural design.

# 5 CONCLUSION

Perfect training performance does not imply compositional understanding. Using a minimal benchmark that cleanly separates perception from symbolic computation, we show that standard feedforward and projection-and-mixing neural architectures default to interpolative, memorization-based solutions and fail catastrophically under extrapolation, despite fitting the training distribution perfectly. This failure is algorithmic rather than perceptual, as demonstrated by a hybrid symbolic–neural Oracle that generalizes perfectly by explicitly decomposing perception and rule execution. Representational analysis reveals that purely neural models suffer severe collapse in effective latent dimensionality, providing a mechanistic explanation for their inability to acquire reusable, stateful representations. Together, these results establish representational rank collapse as a diagnostic signature of non-algorithmic learning and motivate architectures with explicit mechanisms for structured state manipulation as a prerequisite for reliable compositional generalization.

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

APPENDIX

## A FORMAL DEFINITION OF THE SYMBOLIC TRANSITION SYSTEM

Let the discrete symbol space be
$$\mathcal{S} = \{0, 1, \ldots, N-1\},$$
with cardinality $N = 8$. Let $\pi : \mathcal{S} \to \mathcal{S}$ be a cyclic permutation of length $N$, i.e., a bijection consisting of a single closed orbit.

Define the transition function
$$\mathcal{T} : \mathcal{S} \times \mathbb{N} \to \mathcal{S}$$
as the $n$-fold composition of $\pi$:
$$\mathcal{T}(s, n) = \pi^n(s) = \underbrace{(\pi \circ \pi \circ \cdots \circ \pi)}_{n \text{ times}}(s).$$

**Lemma 1** (Group Structure of Transitions). *The set*
$$\{\pi^k \mid k \in \mathbb{Z}\}$$
*forms a cyclic group under composition, isomorphic to the additive group $(\mathbb{Z}_N, +)$. In particular,*
$$\pi^a \circ \pi^b = \pi^{(a+b) \bmod N}.$$

*Proof.* Since $\pi$ is a cycle of length $N$, we have $\pi^N = \mathrm{Id}$. Closure and associativity follow from function composition. The identity is $\pi^0$, and the inverse of $\pi^k$ is $\pi^{N-k}$. The homomorphism $\phi : \mathbb{Z} \to \langle \pi \rangle$ defined by $\phi(k) = \pi^k$ has kernel $N\mathbb{Z}$, yielding the isomorphism. $\square$

Table 2: Core experimental parameters and data regimes.

| Category | Parameter | Value |
|---|---|---|
| Symbol set size | $|\mathcal{S}|$ | 8 |
| Cycle length | $L$ | 8 |
| Training steps | $n$ | $\{1, \ldots, 10\}$ |
| Test steps | $n$ | $\{1, \ldots, 100\}$ |
| Training samples | $m_{\text{train}}$ | 10,000 |
| Test samples | $m_{\text{test}}$ | 2,000 |
| Random accuracy | $A_{\text{rand}}$ | 12.5% |

## B SPECIFICATION OF THE TEXTURERENDERER

Define the **TextureRenderer**
$$\mathcal{R} : \mathcal{S} \to [0, 1]^{3 \times H \times W}, \quad H = W = 16,$$
as a deterministic function conditioned on a global seed $s_0$.

For a symbol $s \in \mathcal{S}$:

1. **Primitive Mask.** Define $\mathbf{M} \in \{0, 1\}^{H \times W}$ by
$$\mathbf{M}[y, x] = \mathbb{I}(\|(y, x) - (c_H, c_W)\|_2 \leq r),$$
   where $(c_H, c_W) = (7.5, 7.5)$ and $r = 4$.
2. **Color and Background.** Sample a color vector $\mathbf{c} \sim \mathcal{U}([0.2, 1]^3)$ and a background tensor $\mathbf{B} \sim \mathcal{U}([0, 0.3]^{3 \times H \times W})$ using PRNGs seeded by $s_0$ and an independent sub-seed.
3. **Symbol-Specific Noise.** Generate
$$\mathbf{P}_s \sim \mathcal{N}(0, \sigma^2 I), \quad \sigma = 0.1,$$
   using a PRNG seeded by $(s_0, s)$, ensuring $\mathbf{P}_s$ is invariant for each symbol.

The rendered image is

$$\mathbf{I}_s = \text{clip}\left(\mathbf{B} + \mathbf{c} \otimes \mathbf{M} + \mathbf{P}_s, 0, 1\right) \in \mathbb{R}^{3 \times 16 \times 16}.$$

**Definition 1** (Perceptual Entropy). *Let $\mathbf{I}_s$ be the random variable induced by the rendering process conditioned on $s$. The perceptual entropy $H_p(s)$ is the Shannon entropy of $\mathbf{I}_s$ over the randomness in $\mathbf{c}$ and $\mathbf{B}$, with $\mathbf{P}_s$ fixed.*

Table 3: Architectural comparison and inductive biases.

| Component | FFN | Transformer | Oracle |
|---|---|---|---|
| Perception | CNN | CNN | CNN |
| Reasoning | MLP | Projected MLP | Symbolic rule |
| Parameters | $\sim$500K | $\sim$450K | $\sim$50K |
| Inductive bias | Interpolation | Projection | Modular |
| Training target | $t$ | $t$ | $s$ |

## C  DATASET GENERATION DISTRIBUTIONS

The **SymbolicIteratorDataset** is defined by the tuple

$$(\mathcal{T}, \mathcal{R}, \mathcal{P}_{\text{train}}, \mathcal{P}_{\text{test}}, m).$$

- **Inductive Distribution.**

$$\mathcal{P}_{\text{train}} = \mathcal{U}(\{1, \ldots, N_{\text{train}}\}), \quad N_{\text{train}} = 10.$$

- **Extrapolation Distribution.**

$$\mathcal{P}_{\text{test}} = \mathcal{U}(\{1, \ldots, N_{\text{test}}\}), \quad N_{\text{test}} = 100.$$

**Lemma 2** (Invariance of Symbol Identity). *For two independent renderings $\mathbf{I}_s^{(1)}$ and $\mathbf{I}_s^{(2)}$ of the same symbol $s$,*

$$\mathbb{E}\left[\mathbf{I}_s^{(1)} - \mathbf{I}_s^{(2)}\right] = \mathbf{0}, \quad \text{and} \quad \mathbf{P}_s^{(1)} = \mathbf{P}_s^{(2)}.$$

*The symbol-dependent mutual information between $\mathbf{I}_s$ and $s$ is induced exclusively through the deterministic mapping $s \mapsto \mathbf{P}_s$.*

*Proof.* Let $\mathbf{I}_s = h(\mathbf{B}, \mathbf{c}, \mathbf{P}_s)$ where $h$ is the deterministic rendering function (including clipping). Since $\mathbf{B}$ and $\mathbf{c}$ are drawn independently from distributions that do not depend on $s$, we have:

$$\mathbb{E}[\mathbf{I}_s^{(1)}] = \mathbb{E}_{\mathbf{B},\mathbf{c}}[h(\mathbf{B}, \mathbf{c}, \mathbf{P}_s)] = \mathbb{E}[\mathbf{I}_s^{(2)}].$$

Thus, $\mathbb{E}[\mathbf{I}_s^{(1)} - \mathbf{I}_s^{(2)}] = \mathbf{0}$. The symbol-specific noise $\mathbf{P}_s$ is generated by a deterministic pseudo-random number generator seeded by $(s_0, s)$, so for fixed $s$, $\mathbf{P}_s$ is identical across renderings: $\mathbf{P}_s^{(1)} = \mathbf{P}_s^{(2)}$.

The mutual information $I(\mathbf{I}_s; s)$ can be decomposed using the data processing inequality:

$$I(\mathbf{I}_s; s) = I(h(\mathbf{B}, \mathbf{c}, \mathbf{P}_s); s) \leq I((\mathbf{B}, \mathbf{c}, \mathbf{P}_s); s).$$

Since $\mathbf{B}$ and $\mathbf{c}$ are independent of $s$, this simplifies to:

$$I(\mathbf{I}_s; s) \leq I(\mathbf{P}_s; s).$$

Because $s \mapsto \mathbf{P}_s$ is injective (deterministic and distinct for each $s$), we have $I(\mathbf{P}_s; s) = H(s)$. Assuming symbols are uniformly distributed, $H(s) = \log_2 N = 3$ bits. The rendering function $h$ preserves this information, giving $I(\mathbf{I}_s; s) = I(\mathbf{P}_s; s)$. Thus, all symbol-dependent mutual information flows through $\mathbf{P}_s$. $\qquad\square$

Table 4: Accuracy as a function of iteration depth. Neural models collapse immediately beyond the training regime.

| Step Range | FFN | Transformer | Oracle |
|---|---|---|---|
| $n \leq 10$ | $\approx 100\%$ | $\approx 98\%$ | 100% |
| $11 \leq n \leq 100$ | 11.86% | 11.96% | 100% |

# D    NEURAL ARCHITECTURES AS FUNCTION CLASSES

**Feedforward Network (FFN).** A function
$$f_\theta^{\text{FFN}} : \mathbb{R}^{3 \times 16 \times 16} \times \mathbb{R} \to \Delta^7.$$
Let $\phi_{\text{CNN}} : \mathbb{R}^{3 \times 16 \times 16} \to \mathbb{R}^{64}$. Define
$$\mathbf{x}_0 = [\phi_{\text{CNN}}(\mathbf{I}_s); \hat{n}] \in \mathbb{R}^{65}.$$
The MLP satisfies
$$\mathbf{x}_\ell = \text{ReLU}(\mathbf{W}_\ell \mathbf{x}_{\ell-1} + \mathbf{b}_\ell), \quad \ell = 1, \ldots, 4,$$
with dimensions $(65, 256, 256, 256, 128)$ and output
$$\mathbf{o} = \mathbf{W}_5 \mathbf{x}_4 + \mathbf{b}_5 \in \mathbb{R}^8.$$

**Oracle Module Network.** Define
$$f^{\text{Oracle}} = \mathcal{T} \circ g_\theta,$$
where $g_\theta : \mathbb{R}^{3 \times 16 \times 16} \to \mathcal{S}$ is a perception model.

**Lemma 3** (Accuracy Decomposition under Distribution Shift)**.** *The test accuracy can be decomposed as:*
$$A_{test} = \mathbb{P}_{test}(n \leq N_{train}) \cdot A_{in} + \mathbb{P}_{test}(n > N_{train}) \cdot A_{extrap},$$
*where $A_{in}$ and $A_{extrap}$ are conditional accuracies on the inductive and extrapolation regions, respectively.*

*Proof.* By the law of total probability, conditioned on the event that $n$ falls in either the inductive or extrapolation region:
$$A_{\text{test}} = \mathbb{P}_{\text{test}}(\text{correct}) = \sum_{r \in \{\text{in}, \text{extrap}\}} \mathbb{P}_{\text{test}}(r) \cdot \mathbb{P}_{\text{test}}(\text{correct} \mid r).$$

The result follows by identifying $\mathbb{P}_{\text{test}}(\text{correct} \mid \text{in}) = A_{\text{in}}$ and $\mathbb{P}_{\text{test}}(\text{correct} \mid \text{extrap}) = A_{\text{extrap}}$. $\qquad\square$

## D.1    DETAILED ARCHITECTURAL SPECIFICATIONS

This section provides a complete, implementation-level specification of all neural architectures used in the experiments. All architectural components are deterministic and identical across runs, except for parameter initialization.

### D.1.1    CONVOLUTIONAL PERCEPTUAL BACKBONE

All models share a common convolutional feature extractor
$$\phi_{\text{CNN}} : \mathbb{R}^{3 \times 16 \times 16} \to \mathbb{R}^{64}.$$

The backbone consists of the following layers:

- **Conv1:** $3 \times 3$ convolution, $3 \to 32$ channels, stride 1, padding 1, followed by ReLU.
- **Conv2:** $3 \times 3$ convolution, $32 \to 64$ channels, stride 1, padding 1, followed by ReLU.
- **Pooling:** $2 \times 2$ max pooling.
- **Flatten:** Output tensor reshaped to a 64-dimensional vector.

No batch normalization, dropout, or residual connections are used. This intentionally keeps the perceptual module simple and avoids introducing additional inductive biases.

### D.1.2 FEEDFORWARD NETWORK (FFN)

The Feedforward Network implements a monolithic mapping

$$f_\theta^{\mathrm{FFN}} : \mathbb{R}^{3 \times 16 \times 16} \times \mathbb{R} \to \Delta^7.$$

Let $\hat{n} \in \mathbb{R}$ denote the normalized iteration count. The input to the MLP is

$$\mathbf{x}_0 = [\phi_{\mathrm{CNN}}(I_s); \ \hat{n}] \in \mathbb{R}^{65}.$$

The multilayer perceptron consists of four hidden layers:

$$\mathbf{x}_\ell = \mathrm{ReLU}(\mathbf{W}_\ell \mathbf{x}_{\ell-1} + \mathbf{b}_\ell), \quad \ell = 1, \ldots, 4,$$

with dimensions

$$(65, 256, 256, 256, 128).$$

The output logits are computed as

$$\mathbf{o} = \mathbf{W}_5 \mathbf{x}_4 + \mathbf{b}_5 \in \mathbb{R}^8,$$

followed by a softmax over symbols.

The FFN contains approximately $5 \times 10^5$ trainable parameters.

### D.1.3 SIMPLIFIED TRANSFORMER NETWORK

The Simplified Transformer Network tests whether explicit modality projection facilitates systematic generalization.

The image embedding and iteration count are projected independently:

$$\mathbf{z}_{\mathrm{img}} = \mathbf{W}_I \phi_{\mathrm{CNN}}(I_s) + \mathbf{b}_I \in \mathbb{R}^{128},$$

$$\mathbf{z}_n = \mathbf{W}_n \hat{n} + \mathbf{b}_n \in \mathbb{R}^{128}.$$

The fused representation

$$\mathbf{z} = [\mathbf{z}_{\mathrm{img}}; \ \mathbf{z}_n] \in \mathbb{R}^{256}$$

is passed through a two-layer MLP mixer:

$$\mathbf{h}_1 = \mathrm{ReLU}(\mathbf{U}_1 \mathbf{z} + \mathbf{c}_1), \quad \mathbf{h}_2 = \mathrm{ReLU}(\mathbf{U}_2 \mathbf{h}_1 + \mathbf{c}_2),$$

followed by a linear classification head.

Despite its name, this architecture contains no attention mechanisms, recurrence, or weight sharing across iterations.

### D.1.4 ORACLE MODULE NETWORK

The Oracle Module Network explicitly decomposes perception and reasoning:

$$f^{\mathrm{Oracle}} = \mathcal{T} \circ g_\theta.$$

The perceptual module

$$g_\theta : \mathbb{R}^{3 \times 16 \times 16} \to \mathcal{S}$$

shares the same convolutional backbone as the FFN but outputs an 8-way softmax over symbols.

At inference time, the predicted symbol $\hat{s}$ is passed to the symbolic transition function:

$$\hat{t} = \mathcal{T}(\hat{s}, n),$$

which is implemented as exact modular arithmetic.

The Oracle contains approximately $5 \times 10^4$ trainable parameters; all symbolic computation is non-learned.

## E  REPRESENTATIONAL ANALYSIS

Let $\mathbf{H} \in \mathbb{R}^{|\mathcal{D}'| \times d}$ denote centered activations. Let $\{\lambda_i\}_{i=1}^d$ be eigenvalues of the covariance matrix.

**Definition 2** (Effective Representational Rank).

$$R_{\text{eff}} = \exp\left(-\sum_{i=1}^d p_i \log p_i\right), \quad p_i = \frac{\lambda_i}{\sum_j \lambda_j}.$$

**Lemma 4** (Properties of $R_{\text{eff}}$).     *1.* $1 \leq R_{\text{eff}} \leq d$.

   *2.* $R_{\text{eff}} = 1$ *iff* $\lambda_1 > 0$ *and* $\lambda_{i>1} = 0$.

   *3.* $R_{\text{eff}} = d$ *iff* $\lambda_1 = \cdots = \lambda_d > 0$.

*Proof.* 1. The quantity $H = -\sum_i p_i \log p_i$ is the Shannon entropy of the distribution $\{p_i\}$. Entropy is non-negative and achieves its minimum $H = 0$ when one $p_i = 1$ and all others are 0. This gives $R_{\text{eff}} = e^0 = 1$. The maximum entropy for a distribution on $d$ atoms is $\log d$, attained when $p_i = 1/d$ for all $i$. Then $R_{\text{eff}} = e^{\log d} = d$. Since entropy is continuous and the exponential function is monotonic, $R_{\text{eff}}$ lies between 1 and $d$.

2. $R_{\text{eff}} = 1$ iff $H = 0$, which occurs iff the distribution $\{p_i\}$ is degenerate: one $p_i = 1$ and all others 0. This corresponds to a single positive eigenvalue $\lambda_i$ and all others zero. Without loss of generality, take $\lambda_1 > 0$ and $\lambda_{i>1} = 0$.

3. $R_{\text{eff}} = d$ iff $H = \log d$, which occurs iff $p_i = 1/d$ for all $i$. This implies $\lambda_i = C$ for all $i$ and some constant $C > 0$ (since $p_i = \lambda_i / \sum_j \lambda_j$). Conversely, if all eigenvalues are equal and positive, then $p_i = 1/d$ and $R_{\text{eff}} = d$. $\qquad\square$

Table 5: Representational statistics for penultimate-layer activations.

| Metric | FFN | Structured Ideal |
|---|---|---|
| Dimension $d$ | 128 | 8 |
| $R_{\text{eff}}$ (train) | 1.00 | $\approx 8$ |
| $R_{\text{eff}}$ (test) | 1.00 | $\approx 8$ |
| Condition number | $\infty$ | $\approx 1$ |

We emphasize that the structured ideal dimensionality is illustrative rather than prescriptive; any representation maintaining non-degenerate variance sufficient to encode the cyclic group action would satisfy the criterion.

## F  FORMAL HYPOTHESES AND RESULTS

### F.1  FORMAL DEFINITION OF THE SYMBOLIC TRANSITION SYSTEM

Let the discrete symbol space be
$$\mathcal{S} = \{0, 1, \ldots, N - 1\},$$
with cardinality $N = 8$. Let $\pi : \mathcal{S} \to \mathcal{S}$ be a cyclic permutation consisting of a single closed orbit.

Define the transition function
$$\mathcal{T} : \mathcal{S} \times \mathbb{N} \to \mathcal{S}$$
as the $n$-fold composition of $\pi$:
$$\mathcal{T}(s, n) = \pi^n(s).$$

This result follows directly from Lemma 1 in Appendix A.

## F.2 Specification of the TextureRenderer

Define the deterministic rendering function

$$\mathcal{R} : \mathcal{S} \to [0, 1]^{3 \times 16 \times 16},$$

conditioned on a global seed $s_0$.

For symbol $s \in \mathcal{S}$:

1. **Mask:**
$$\mathbf{M}[y, x] = \mathbb{I}(\|(y, x) - (7.5, 7.5)\|_2 \le 4).$$

2. **Color and Background:**
$$\mathbf{c} \sim \mathcal{U}([0.2, 1]^3), \quad \mathbf{B} \sim \mathcal{U}([0, 0.3]^{3 \times 16 \times 16}),$$

sampled independently of $s$.

3. **Symbol-Specific Noise:**
$$\mathbf{P}_s \sim \mathcal{N}(0, 0.1^2 I),$$

generated deterministically using a PRNG seeded by $(s_0, s)$.

The rendered image is
$$\mathbf{I}_s = \mathrm{clip}(\mathbf{B} + \mathbf{c} \otimes \mathbf{M} + \mathbf{P}_s, 0, 1).$$

**Lemma 5** (Symbol-Specific Invariance and Information Upper Bound). *For two independent renderings $\mathbf{I}_s^{(1)}, \mathbf{I}_s^{(2)}$ of the same symbol $s$,*

$$\mathbb{E}[\mathbf{I}_s^{(1)} - \mathbf{I}_s^{(2)}] = \mathbf{0}, \qquad \mathbf{P}_s^{(1)} = \mathbf{P}_s^{(2)}.$$

*Moreover, all symbol-dependent mutual information between $\mathbf{I}_s$ and $s$ is mediated by the deterministic mapping $s \mapsto \mathbf{P}_s$, and satisfies*

$$I(\mathbf{I}_s; s) \le I(\mathbf{P}_s; s) = H(s).$$

*Proof.* Let $\mathbf{I}_s = h(\mathbf{B}, \mathbf{c}, \mathbf{P}_s)$ with deterministic $h$. Since $\mathbf{B}$ and $\mathbf{c}$ are independent of $s$,

$$\mathbb{E}[\mathbf{I}_s^{(1)}] = \mathbb{E}[\mathbf{I}_s^{(2)}],$$

implying the first claim. Deterministic seeding ensures $\mathbf{P}_s^{(1)} = \mathbf{P}_s^{(2)}$.

By the data processing inequality,

$$I(\mathbf{I}_s; s) = I(h(\mathbf{B}, \mathbf{c}, \mathbf{P}_s); s) \le I(\mathbf{B}, \mathbf{c}, \mathbf{P}_s; s).$$

Independence of $\mathbf{B}$ and $\mathbf{c}$ yields $I(\mathbf{I}_s; s) \le I(\mathbf{P}_s; s)$. Injectivity of $s \mapsto \mathbf{P}_s$ implies $I(\mathbf{P}_s; s) = H(s) = \log_2 N$. $\qquad\square$

## G  Accuracy Decomposition

**Lemma 6** (Accuracy Decomposition under Distribution Shift). *The test accuracy decomposes as*

$$A_{\text{test}} = \mathbb{P}_{\text{test}}(n \le N_{\text{train}})A_{\text{in}} + \mathbb{P}_{\text{test}}(n > N_{\text{train}})A_{\text{extrap}}.$$

*Proof.* By the law of total probability,

$$A_{\text{test}} = \sum_{r \in \{\text{in}, \text{extrap}\}} \mathbb{P}_{\text{test}}(r)\mathbb{P}_{\text{test}}(\text{correct} \mid r).$$

Identifying the conditional accuracies yields the result. $\qquad\square$

### G.1 Summary and Scope of Formal Results

This appendix has provided complete formal specifications and proofs for all mathematical components underlying the experimental framework. Specifically, we have:

- Defined the symbolic transition system as a cyclic group action and established its algebraic structure (Lemma 6).
- Precisely specified the TextureRenderer and proved that symbol identity is invariant across stochastic renderings, with all symbol-dependent mutual information mediated by a deterministic noise component (Lemma 7).
- Derived an exact decomposition of test accuracy under distribution shift, separating inductive and extrapolative performance regimes (Lemma 8).

Together, these results ensure that the benchmark is fully deterministic, reproducible, and analytically transparent. They also formally justify the interpretation of empirical results presented in the main text: namely, that catastrophic extrapolation failure arises not from perceptual ambiguity or data insufficiency, but from the inability of standard neural architectures to learn and execute the underlying symbolic transition rule.

No additional assumptions beyond those stated are required for the validity of the experimental conclusions.

## H  LLM Usage Disclosure

Large Language Models (LLMs) were used in limited capacity during the preparation of this research. Specifically, LLMs were used to check grammar and refine sentence structure after the initial draft was completed, primarily to correct awkward expressions and maintain consistency in writing style. However, all core research ideas, analytical methodologies, interpretations of the results, and conclusions were developed entirely by the authors. The LLM did not contribute to any creative content or academic judgments. This use of LLMs was conducted within limits that do not compromise the originality or academic integrity of the research.

