# OpenReview forum: "COMPOSITIONAL CHOKE POINTS: EVIDENCE OF EMERGENT RANK COLLAPSE LIMITING ITERATIVE GENERALIZATION"
_ICLR.cc/2026/Workshop/Sci4DL — Submitted to Sci4DL 2026_

### Official Review · Reviewer_RxQj · 2026-02-26

**Fit:** 1
**Significance:** 1
**Confidence:** 3

**Summary:**

This paper looks at learning a target function which maps a hidden state (disguised as an image) and integer $n$ through a permutation that is iterated $n$ times to produce the target. An oracle model that knows the underlying permutation structure is able to perfectly learn the task, but two neural networks are unable to do so or at least fail to extrapolate beyond the values of $n$ used for training.

**Strengths:**

The idea of learning the composition of permutation functions is interesting if it demonstrates a limit in neural network learning. The introduction is readable and motivates the problem well.

**Suggestions:**

I found the results provided very minimal evidence for the many claims made in the paper. My interpretation of the results were that the authors just couldn't get these architectures to work for the problem, not that the problem is inherently intractable to ANNs. For instance, if the variable n were encoded as cos(an), sin(bn) then presumably a network could learn a and b and downstream weights to correctly infer the period-8 dependence on $n$ and perfectly extrapolate.

If the claims of the paper hold, then it should be possible to remove the image mapping part and just try to predict $t$ from $(s, n)$ and show the same inability to learn and effects. Also, I could imagine that the results shown in the paper are sensitive to what range of $n$ is available in training. You used 1-10, but what if you use 1-16, 1-24, or more? That way the network actually sees the periodicity. That would give it more of a chance to extrapolate.

Claims that I thought were confusing and not well-supported were:
* line 169: "preceptual abstraction is fully solvable by a standard CNN"; it's unclear what perceptual abstraction even means
* line 182: "compositional choke point"

The "simplified transformer" appears to be another feedforward MLP plus conv layer in disguise (compare D1.2 and D1.3, the equations are the same since you can combine the weights W and U in the "transformer" to get a linear layer as in the FFN; the only difference appears to be width). I would suggest using an actual transformer. The claim the "transformer" does something different in the latent space than the FFN isn't supported by evidence here.

Smaller points that could be improved in the paper:
* It should be discussed whether recurrence could help some of these problems. I could imagine the right recurrent network could learn a periodic orbit that allows the parameter n to exhibit periodic behavior.
* The actual mapping $\pi^{(n)}(s)$ should be explained in the main body rather than the appendix. The action of the superscript $n$ isn't clear in the main paper. I would avoid using as much technical group theory language. You can just set $S = \{0, 1, \ldots, 7\}$ for clarity.
* Similarly, some basic description of the texture map $\mathcal{R}$ should be included in the main body. An example image would help.
* Fig 1 is very hard to read. Fonts are too small.
* line 136: clarify "extreme variance and poor conditioning", similarly clarify effective rank in main text
* lemmas in the appendix: I don't think these add much value. I am unclear what Lemma 2 means and why we should care; it isn't explained. Lemma 3 is just saying the error/accuracy is an average of different n's. This is nothing new and doesn't seem to merit a proof.
* appendix symbol $\Delta$ appearing as $\Delta^7$ doesn't seem to be defined
* appendix section F repeats what was given earlier
* I could tell from the writing style that an LLM was used to write/revise some of the work. While the authors disclose this, I found that overall there was a lot of flowery technical language that could be simplified to make things more understandable.

---

### Official Review · Reviewer_kbzx · 2026-02-26

**Fit:** 2
**Significance:** 1
**Confidence:** 2

**Summary:**

The paper constructs a benchmark where models must learn a cyclic permutation rule πⁿ(s) from stochastically rendered images. Models are trained on iteration depths n ∈ {1,…,10} and tested on n ∈ {1,…,100}. A feedforward network and a “Simplified Transformer” achieve near-perfect training accuracy but collapse to chance on extrapolation, while a hybrid Oracle that hardcodes symbolic reasoning generalizes perfectly. The paper analyzes penultimate-layer representations and finds effective rank collapse, which it proposes as a diagnostic for memorization-based rather than algorithmic learning.

**Strengths:**

- The experimental design sort of separates perception from reasoning and might become a useful framework.
- The appendices are thorough, providing proofs of group structure, information-theoretic analysis of the renderer, and full architectural specifications.

**Suggestions:**

- The core finding is not novel. That FFNs and CNNs cannot extrapolate beyond their training distribution is extensively documented. This paper adds another benchmark demonstrating the same known failure, but does not advance our understanding of why it occurs or how to fix it.
- The rank collapse observation is interesting but not causal. Does rank collapse cause extrapolation failure, or is it just a byproduct of memorization on a tiny discrete input space?
- The training setup makes memorization the rational strategy. With only 10 discrete values of n in training and 80 total (s, n) → t mappings, it's easy to memorize the look-up table. The model might simply have no motivation to develop any iterative strategies.

---

### Meta-Review · Area_Chair_yBfF · 2026-02-28

**Recommendation:** Reject

**Metareview:**

This is a cool idea. I actually like the approach of constructing nice datasets that show clean separations in behavior and dissecting the models post hoc to understand what's going on. Unfortunately, this paper has some problems. In addition to those identified by the reviewers, I'd note that the oracle model seems overpowered: seemed that you're only training the perception module, with the permutation implemented deterministically after that? It's not clear to me what we're learning about real nets by comparing to that.

I'd encourage taking another crack at this with a more careful setup and more intentional empirics.

---

### Decision · Program_Chairs · 2026-03-02

Reject